# Rhabdomyolysis related to acute recreational drug toxicity—A Euro-DEN study

**Wojciech Waldman** [1,2], **Piotr M. Kabata** [2], **Alison M. Dines**[3], **David M. Wood**[3,4],
**Christopher Yates**[5], **Fridtjof Heyerdahl**[6], **Knut Erik Hovda**[6], **Isabelle Giraudon**[7], **Euro-DEN Research Group**[¶], **Paul I. Dargan**[3,4], **Jacek Sein Anand** [1,2]

1 Department of Clinical Toxicology, Medical University of Gdansk, Gdansk, Poland, 2 Pomeranian Centre of Toxicology, Gdansk, Poland, 3 Clinical Toxicology, Guy's and St Thomas' NHS Foundation Trust and King's Health Partners, London, United Kingdom, 4 Faculty of Life Sciences and Medicine, King's College London, London, United Kingdom, 5 Hospital Universitari Son Espases, Palma de Mallorca, Spain, 6 Medical Division, Oslo University Hospital, Oslo, Norway, 7 European Monitoring Centre for Drugs and Drug Addiction (EMCDDA), Lisbon, Portugal

¶ Euro-DEN Research Group

**Data Availability Statement:** The dataset contains date and time of patients admission to the hospital, which could potentially identify a patient;

## Abstract

### Background

This study was conducted to retrospectively assess the relationships between: rhabdomyolysis (quantified by creatine kinase (CK) activity) and kidney injury (quantified by serum creatinine concentration), sex, age, body temperature on admission, presence of seizures, and agitation or aggression in patients presenting to the Emergency Department with acute recreational drug toxicity. We also investigated the association with the substances ingested.

### Methods

All presentations to the 16 sentinel Euro-DEN centres in 10 European countries with acute recreational drug toxicity during the first year of the Euro-DEN study (October 2013 to September 2014) were considered. Cases that had abnormal CK activity recorded as part of routine clinical care were divided into 3 cohorts depending on peak CK activity. Cases with normal CK activity were included as a control group (4th cohort).

### Results

Only 1,015 (18.4%) of the 5,529 Euro-DEN presentations had CK activity concentration recorded. Of this group 353 (34.8%) had also creatinine concentration measured. There were 375 (36.9%) with minor rhabdomyolysis, 69 (6.8%) with moderate rhabdomyolysis, and 24 (2.4%) with severe rhabdomyolysis; 547 (53.9%) were included in the control group. There was a positive correlation between CK activity and creatinine concentration (correlation coefficient r = 0.71, p<0.0001). There was no correlation between CK activity and body temperature at the time of presentation to the ED (correlation coefficient r = 0.07, p = 0.03). There was a positive correlation between CK activity and length of stay in the hospital (r = 0.31, p<0.001). There was no association between CK activity and the presence of seizures (p = 0.33) or agitation/aggression (p = 0.45), patients age (p = 0.4) or sex (p = 0.25). The 5

furthermore recreational drug use is a sensitive area for many individuals and therefore the risks associated with identification are greater in this context. The data can be obtained from European Drug Emergencies Network steering committee for researchers who meet the criteria for access to confidential data. The restrictions have been imposed by the EuroDEN steering committee. For inquiries regarding data access, please contact Mrs Barbara Whetton, phone number: +44 (0)20 7188 5848 barbara.whetton@gstt.nhs.uk.

**Funding:** With financial support from the DPIP/ISEC Programme of the European Union and the European Monitoring Centre for Drugs and Drug Addiction. All the authors had funding from the European Commission through the Euro-DEN project except ML and EL, whose costs were co-funded by the Swiss Centre for Applied Human Toxicology (SCAHT) and KP, IG and RS.

**Competing interests:** The authors have declared that no competing interests exist.

**Abbreviations:** AKI, acute kidney injury; ALT, alanine aminotransferase; AST, aspartate aminotransferase; CK, creatine kinase; ED, emergency department; Euro-DEN, European Drug Emergencies Network; GBL, γ-butyrolactone; GHB, γ-hydroxybutyric acid; IQR, interquartile range; KDIGO, Kidney Disease: Improving Global Outcomes; LOS, Length of stay in hospital; PSS, Poisoning Severity Score.

most common agents amongst patients presenting with rhabdomyolysis were: cocaine (n = 107; 22.9% presentations), amphetamine (76; 16.2%), cannabis (74; 15.8%), GHB/GBL (72; 15.4%) and heroin (67; 14.3%). The distribution of rhabdomyolysis in 5 most common drugs was (drug; patients with rhabdomyolysis, patients without rhabdomyolysis): cocaine (107, 122), cannabis (74, 117), GHB/GBL (72, 81), amphetamine (76, 66), heroin (67, 70).

## Conclusions

Abnormal values of CK activity occurred in almost half (46.1%) of presentations to the Emergency Department with acute recreational drug toxicity in whom CK activity was measured; however, severe rhabdomyolysis is seen in only a small minority (2.4%). Those with rhabdomyolysis are at significantly higher risk of kidney injury and have a longer length of hospital stay.

## Introduction

Rhabdomyolysis is a syndrome that is characterized by the disintegration of striated muscle and the leakage of muscle-cell contents including myoglobin and creatine kinase (CK). The first detailed reports of rhabdomyolysis, related to the crush syndrome, and complicated with acute renal failure, were presented by Bywaters and Beall in the 1940s [1,2]. CK is more reliable than myoglobin in assessing the presence and intensity of damage to the muscles because overall CK degradation and removal are slow, the concentration of CK remains elevated much longer and in a more consistent manner than that of myoglobin. Definitions of rhabdomyolysis vary from that requiring CK elevation greater than approximately five times the upper limit of normal to more recent definitions requiring CK elevation greater than 50 times the upper limit of normal with renal insufficiency [3]. Rhabdomyolysis is a serious condition, which may lead to life threatening complications.

The causes of rhabdomyolysis can be divided into hereditary and acquired ones. The acquired causes are classified as traumatic and non-traumatic. Non-traumatic causes are the most common during peacetime and include psychoactive drugs, alcohol abuse, and many others. The medical literature is filled with case reports of rhabdomyolysis resulting from snake and spider bites [4–6] to drugs like cocaine, methamphetamine [7–11], 3,4-methylenedioxymethamphetamine (MDMA) [12,13] and plant toxins [14].

Despite the fact that at least 150 medications and toxins have been described which may lead to rhabdomyolysis, recreational drugs and alcohol are the most common causes [15]. Many factors are known to contribute to development of rhabdomyolysis including seizures, agitation or aggression, immobilisation, excessive muscle activity, hypo- or hyperthermia [3,15–20], and potentially also direct substance-associated toxic effects in susceptible persons [21]. There have been no large, multicentre studies on the incidence of rhabdomyolysis in acute recreational drug toxicity.

The European Drug Emergencies Network (Euro-DEN) project collects data on presentations to sentinel Emergency Departments (ED) in Europe in whom the primary reason for presentation is acute recreational drug toxicity [22,23]. During the first year from October 2013 to September 2014 data was collected from 16 sentinel centres in 10 European countries. The aim of this study was to use data from the Euro-DEN project to determine the relationship between maximum CK activity and factors frequently associated with rhabdomyolysis

(creatinine concentration as a marker of acute kidney injury (AKI), length of stay in hospital, patients' temperature at admission, presence of seizures, agitation or aggression, hyperthermia) as well as others that might influence it (substances used by the patient and their number in case of multisubstance abuse).

## Materials and methods

### Ethics approval and consent to participate

The study involved retrospective analysis of anonymised clinical data records. The need for approval was waived by the ethical committees of respective centres participating in the study.

### Data collection

All acute recreational drug toxicity presentations to the 16 sentinel ED in 10 European countries participating in the first year (October 2013 to September 2014) of the Euro-DEN project were included retrospectively using standard Euro-DEN methodology [22]. Patients presenting with other main complaints (including abstinence syndromes) who were under influence of psychoactive substances were excluded. Presentations involving intoxication with ethanol without other coingested psychoactive substances was excluded from the study. In case of lack of information regarding used substance from patient history or laboratory results, collective label "unknown substance" was used.

After completing first year of data collection, gathered dataset was revised and some of the variables (including creatinine concentration and CK activity) were removed. As a result, EuroDEN and EuroDEN plus data contain only data from October 2013 to September 2014.

In the Euro-DEN dataset only highest recorded CK activity and serum creatinine concentrations measured during patients stay were recorded [22,23].

### Data analysis

The dataset was converted from the Euro-DEN Excel spreadsheet to a comma separated values file. Data wrangling, analysis and visualisations for this analysis were performed in R programming language for statistics [24] with use of modules: ggplot2 [25], dplyr [26], gridExtra [27], reshape2 [28]. Analysis was performed by the Euro-DEN centre located in Gdansk Poland.

Only 1015 (18.4%) of the 5,529 Euro-DEN presentations over the study period had CK activity recorded in the database and included in the analysis. Parameters analysed were: time from use to presentation, self-reported number of agents used, what agents were used (reported), presence of seizures, body temperature on admission, presence of agitation/aggression, maximum CK activity recorded, maximum creatinine recorded, length of stay in hospital (LOS). All the missing values were filled with 'NA' value.

For further analysis presentations were divided into 4 cohorts depending on maximum CK activity. The values used for the partitioning of the data were based on values used in Poison Severity Score (PSS) [29]: 1. no rhabdomyolysis (CK below 250 IU/L), 2. minor rhabdomyolysis (CK 251–1,500 IU/L), 3. moderate rhabdomyolysis (CK 1,501–10,000 IU/L), 4. severe rhabdomyolysis (CK above 10,000 IU/L). It should be noted, that the severity of rhabdomyolysis does not have to be consistent with severity of poisoning, which was not assessed in this analysis.

Fragmented groups of substances sharing chemical and toxicological properties (for example benzodiazepines) were treated as one causative agent during the analysis.

Where kidney function was analysed, we have taken creatinine concentration of 1.2 mg/dl or higher, to be considered as a marker of kidney injury. This was a result of study design, which recorded only the highest creatinine concentration during patient stay at the hospital.

## Statistical analysis

Threshold for statistical significance for the testing was established at $p < 0.01$ (probability of type I error smaller than 1%). We used both range and interquartile range (IQR) as measures of data spread. Correlations were tested using Spearman's rho. Where analysis of means between the groups was needed, Welch Two Sample t-test was performed.

## Results

Only presentations with CK activity recorded (N = 1015) were included in the analysis. The median (IQR; range) CK was 228 (124.5–448.0; 30–169,700) IU/L.

As shown in Table 1, 46.1% of this cohort had a CK > 250 IU/L, the majority of whom had minor rhabdomyolysis.

### CK vs. creatinine

Of the 1015 patients included in our analysis only 353 (39.1%) had both CK and creatinine concentration recorded.

In the total studied population, median creatinine was 0.90 mg/dl (min 0.40 mg/dl, max 6.94 mg/dl, IQR 0.72–1.10 mg/dl). For patients with no rhabdomyolysis median creatinine was 0.80 mg/dl (min 0.41 mg/dl, max 1.47 mg/dl, IQR 0.69–0.91 mg/dl). For patients with rhabdomyolysis median creatinine was 0.99 mg/dl (min 0.40 mg/dl, max 6.94 mg/dl, IQR 0.81–1.20 mg/dl).

Median creatinine of patients with minor rhabdomyolysis was 0.91 mg/dl (min 0.40 mg/dl, max 1.75 mg/dl, IQR 0.80–1.10 mg/dl). Median creatinine of patients with moderate rhabdomyolysis was 1.01 mg/dl (min 0.56 mg/dl, max 2.40 mg/dl, IQR 0.86–1.37 mg/dl). Median creatinine of patients with severe rhabdomyolysis was 1.65 mg/dl (min 0.54 mg/dl, max 6.94 mg/dl, IQR 0.95–2.97 mg/dl). The correlation between CK activity and creatinine was 0.71, $p < 0.0001$. In the patients with both creatinine and CK measured 44 (12.5%) had creatinine concentration of 1.2 mg/dl or higher, which was considered a marker of kidney injury.

The prevalence of kidney injury is presented in Table 2. The presence of rhabdomyolysis increased the odds of kidney injury 3.97 times.

The relationships between peak CK activity and peak creatinine concentration are shown in Fig 1; as shown, there was a significant positive correlation between peak CK activity and creatinine concentration.

**Table 1. Creatine kinase (CK) results in the 1015 Euro-DEN presentations.**

| CK (IU/L) | ≤ 250 IU/l | 250–1500 IU/l | 1501–10000 IU/l | CK >10000 IU/l | Total |
|---|---|---|---|---|---|
| n | 547 (53.9%) | 375 (36.9%) | 69 (6.8%) | 24 (2.4%) | 1015 (100%) |
| Min | 30 | 252 | 1522 | 10036 | 30 |
| Max | 250 | 1397 | 9441 | 169700 | 169700 |
| Median | 132.0 | 441.0 | 2595.0 | 21934.0 | 228.0 |
| IQR | 94.0–183.0 | 319.0–684.5 | 1986.0–4030.0 | 16760.0–63640.0 | 124.0–488.0 |

**Table 2. Incidence of kidney injury in studied cohorts.**

| Rhabdomyolysis Cohort | [ALL] N = 353 | Did the patient develop kidney injury? | | Odds Ratio | p for odds ratio | p overall |
|---|---|---|---|---|---|---|
| | | No N = 309 | Yes N = 44 | | | |
| NONE | 188 (53.3%) | 177 (57.3%) | 11 (25.0%) | Reference | Reference | <0.001 |
| MINOR | 124 (35.1%) | 109 (35.3%) | 15 (34.1%) | 2.20 [0.97;5.13] | 0.06 | |
| MODERATE | 31 (8.78%) | 19 (6.15%) | 12 (27.3%) | 9.95 [3.84;26.4] | <0.001 | |
| SEVERE | 10 (2.83%) | 4 (1.29%) | 6 (13.6%) | 22.9 [5.57;106] | <0.001 | |
| All patients who developed rhabdomyolysis | 165 (46.7%) | 132 (42.7%) | 33 (75.0%) | 3.97 [1.99;8.55] | <0.001 | <0.001 |

### CK vs age

The youngest patient in the study was 12 years old, the oldest was 88 years old. Median age of a patient was 30 years.

There was no significant correlation between the age of a patient and CK activity (r = 0.03, p = 0.18).

### CK vs sex of a patient

The study included 241 women and 774 men. Rhabdomyolysis was observed in 68 (28,2%) women and 400 (51,7%) of male patients. The differences in CK activity were significant only for patients with no rhabdomyolysis (p<0.01).

### CK vs length of stay in hospital

The length of stay in hospital (LOS) was longer in those with more severe rhabdomyolysis. In the total studied population, median LOS was 12.75 h (min 0.58 h, max 664 h, IQR 5.00–37.58

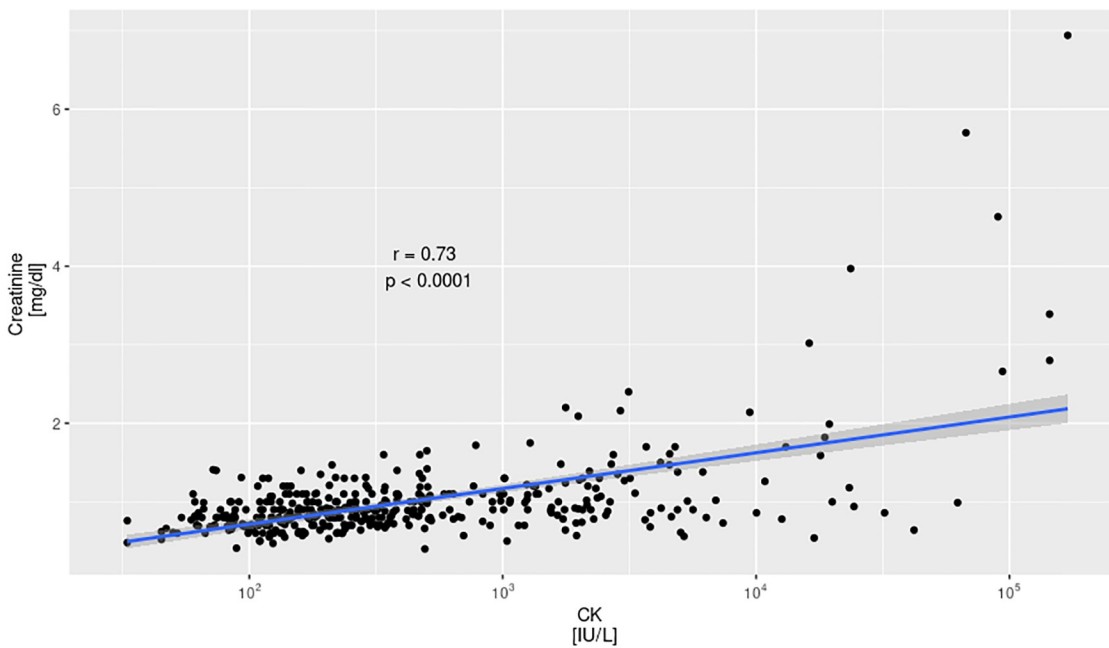

**Fig 1. Correlation between CK activity and max serum creatinine concentrations.** r—Pearson correlation coefficient, p—confidence level.

h). For patients with no rhabdomyolysis median LOS was 10.23 h (min 0.62 h, max 501 h, IQR 3.97–27.37 h). Median LOS of patients with minor rhabdomyolysis was 9.92 h (min 0.98 h, max 453.78 h, IQR 4.20–25.87 h). Median LOS of patients with moderate rhabdomyolysis was 25.25 h (min 0.58 h, max 664.00 h, IQR 12.63–55.91 h). Median LOS of patients with severe rhabdomyolysis was 108.98 h (min 2.81 h, max 661.15 h, IQR 43.09–255.80 h). The correlation between CK activity and LOS was 0.31, p<0,001.

## CK vs. body temperature on admission

Of 1015 patients included in our analysis body temperature on admission was measured in 933 (91.92%). The data indicates that there is no correlation between body temperature on admission and rhabdomyolysis (r = 0.07, p = 0.03).

## CK vs. presence of seizures and agitation or aggression

There was a no significant relationship between seizures being present at any time during the presentation and the severity of the rhabdomyolysis. (Table 3).

## CK vs. drugs used

As shown in Table 4 the drugs used by those with rhabdomyolysis were similar to those without rhabdomyolysis.

There was no significant correlation between those values of CK activity in studied cohorts, and the number of agents (self-reported or tested) taken by the patient.

We have calculated the odds ratios for most frequently used substances. The calculation show, that the highest potential for rhabdomyolysis development is seen in amphetamine (OR 1.41).

**Table 3. CK activity, seizures at any time during the admission and agitation/aggression on admission.** For 2 patients (1 in No rhabdomyolysis and 1 in Severe rhabdomyolysis cohort) there was no information regarding presence of seizures. For 1 patient (in No rhabdomyolysis cohort) there was no information regarding presence of agitation/aggression.

| | Total | | No rhabdomyolysis | | Mild rhabdomyolysis | | Moderate rhabdomyolysis | | Severe rhabdomyolysis | |
|---|---|---|---|---|---|---|---|---|---|---|
| **Seizures** | **Yes** | **No** | **Yes** | **No** | **Yes** | **No** | **Yes** | **No** | **Yes** | **No** |
| n (% of patients in cohort) | 72 (7.1%) | 941 (92.7%) | 34 (6.2%) | 512 (93.6%) | 28 (7.5%) | 347 (92.5%) | 5 (7.3%) | 64 (92.8%) | 5 (20.8%) | 18 (75.0%) |
| Median CK | 266 | 225 | 135 | 131 | 483 | 440 | 2552 | 2624 | 22080 | 19450 |
| Agitation / Aggression | Yes | No | Yes | No | Yes | No | Yes | No | Yes | No |
| n (% of patients in cohort) | 390 (38.4%) | 624 (61.5%) | 176 (32.2%) | 370 (67.6%) | 165 (44.0%) | 210 (56.0%) | 37 (53.6%) | 32 (46.4%) | 12 (50.0%) | 12 (50.0%) |
| Median CK | 286 | 208 | 144 | 125 | 469 | 430 | 2365 | 2834 | 21380 | 22440 |

**Table 4. 5 Most commonly reported psychoactive substances (as stated in medical records).** For a reference number of unidentified agents was added.

| Total population | | Patients without rhabdomyolysis | Patients with rhabdomyolysis | Odds ratio | Odds ratio Confidence interval |
|---|---|---|---|---|---|
| **Substance used** | **Frequency (N, %)** | **Frequency (N, %)** | **Frequency (N, %)** | | |
| Cocaine | 229 (22.6%) | 122 (22.3%) | 107 (22.9%) | 1.03 | 0.79–1.34 |
| Cannabis | 192 (18.9%) | 117 (21.4%) | 74 (15.8%)76 (16.2%) | 0.74 | 0.54–0.99 |
| GHB/GBL | 153 (15.1%) | 81 (14.8%) | 72 (15.4%) | 1.04 | 0.75–1.44 |
| Amphetamine | 142 (14.0%) | 66 (11.7%) | 76 (16.2%) | 1.34 | 0.97–1.89 |
| Heroin | 137 (13.5%) | 70 (12.1%) | 67 (14.3%) | 1.12 | 0.8–1.58 |
| Unknown substance | 45 (4.3%) | 21 (3.8%) | 24 (5.1%) | | |

## Discussion

Our results have shown that in almost half of the patients presenting to the ED with acute recreational drug toxicity in whom clinicians decided to measure CK activity due to higher risk of rhabdomyolysis, had elevated CK activities. This result might be biased by the fact that probably some of the centres measured CK activity only in patients that either clinically showed signs of muscle injury or their history suggested possible development of the condition. The majority of these had minor or moderate rhabdomyolysis, and only 2.4% developed severe rhabdomyolysis. There was a positive association between creatinine concentrations and the severity of rhabdomyolysis. This is a known phenomenon, since both CK and creatinine are products of muscle damage. However, the median creatinine concentration was elevated only in patients with CK activities >10 000 U/L. Due to study design, only highest creatinine concentration was recorded, thus it was not possible to check the patients against Kidney Disease: Improving Global Outcomes (KDIGO) acute kidney injury criteria. Data were also not available on the need for renal replacement therapy. The mechanism of rhabdomyolysis in acute recreational drug toxicity is variable—some drugs have direct myocytotoxic effects, whilst others can increase physical activity to deleterious levels, produce ischemia due to arterial vasoconstriction and precipitate seizures or hyperthermia [21,30–35]. Depressant intoxication can also lead to immobilisation and rhabdomyolysis related to compression and ischemic injury. For example, in a small previous study CK levels were correlated with the duration of coma and coma was more prolonged in patients with combined use of γ-hydroxybutyric acid (GHB) and stimulants [33,34]. Our study shows that GHB and stimulants are among the most prevalent causes of rhabdomyolysis.

Although illicit drugs are well-described precipitants of rhabdomyolysis [36,37], there is limited data on the frequency of rhabdomyolysis in patients presenting with acute recreational drug toxicity.

Due to different study methodology, inclusion criteria based upon emergency or hospital physician coding, and variable definitions of "renal failure", these studies may be difficult to compare. Importantly none of studies included patients discharged from ED or performed follow-up of patients to identify adverse events post-hospital discharge. We have found only one study, presented by Grunau et al., that compared CK activities of patients who were admitted to the hospital to those measured in all ED presentations. In this study, 400 cases of rhabdomyolysis, defined as CK level greater than 1 000 U/l, were found by the authors in 9,509 patients who had CK activity test ordered out of 235,947 ED visits. 30-day follow-up was performed after ED treatment. 35% of patients were discharged home from ED. The most common ED discharge diagnoses were related to recreational drug use, infections, and traumatic or musculoskeletal complains. Within 30 days 21 (5.3%) patients died with 18 (4.5%) requiring hemodialysis. AKI occurred in 151 (38%) patients. In this work, the authors found that higher CK values were not associated with worse outcomes and concluded that initial creatinine was the best predictor of outcome [38]. Our data show that very high activities of CK can be used as a predictor of prolonged hospitalisation.

As there are no universal CK activity criteria to establish diagnosis of rhabdomyolysis, we used the Poisoning Severity Score to determine severity of rhabdomyloysis. Partitioning to cohorts was important for the analysis, due to data spread (maximum CK activity was 5657 times higher than the minimum). Cohorts with normal CK activity vs. elevated were similar in size (N = 547 vs. N = 468). A group with elevated CK represented 8.5% of total and 46.2% of patients where CK was measured. Only 353 patients had both CK and creatinine measured.

Literature data show that higher CK activities correlated with degree of muscle injury, but correlated only slightly with the development of AKI or mortality [15,36,39–45]. The incidence

of AKI secondary to rhabdomyolysis varies from 10% to 59%, whilst it is estimated that 5% to 15% of AKI cases can be attributed to rhabdomyolysis [15,16,36,39,41–43,46]. In previous studies on cocaine-intoxicated patients, the prevalence of AKI as the consequence of rhabdomyolysis was reported as 24% to 33% [9,11]. Based on our analysis we have observed a statistically significant positive correlation between highest CK activity and highest creatinine concentration in a general population as well as in most of the cohorts. The severity of rhabdomyolysis was associated with longer hospitalisation and thus could lead to higher treatment costs.

Our study did not show any correlation between the number of substances involved and rhabdomyolysis. The most common agents used by patients who developed rhabdomyolysis, were the same as those in patients without rhabdomyolysis with the 5 most common being cocaine, cannabis, GHB/GBL (γ-butyrolactone) amphetamine and heroine. Cocaine is a known factor contributing to development of muscular injury [9,11,46]. A PubMed search for "cannabis+rhabdomyolysis" showed only 8 results, of which only one was a report of connection between cannabis use and incidence of rhabdomyolysis, however the direct cause of the injury was falling asleep while sitting cross-legged. In our data, only 12 patients having seizures have used cannabis. Of this group 9 had no rhabdomyolysis and 3 minor rhabdomyolysis, which supports that observation. There is only a limited data on relationship between GHB/GBL and rhabdomyolysis in the literature, and most were reported cases involving muscle injury during withdrawal syndrome [33,34,47]. Considering the clinical effect of GHB/GBL, it can be assumed that rhabdomyolysis is caused by patient's immobility for prolonged periods. It should be noted, that MDMA was the most prevalent substance found in patients who developed severe rhabdomyolysis, however this group was relatively small, which might have influenced the result.

## Limitations of the study

The study was designed as a retrospective analysis of a large volume of clinical history records.

Laboratory test data (CK activity and creatinine) were only recorded if these were taken as part of routine clinical care; this could potentially lead to a selection bias in the studied data as the patients with clinical signs of rhabdomyolysis would be tested more frequently. No data regarding physical activity in days before presentation was gathered, opening possibility to including patients who developed exertional rhabdomyolysis in the group. Due to the study design only the highest recorded CK activity and creatinine concentration was included, and data were not collected on previous renal function to be able to confirm that any kidney injury present was AKI rather than underlying chronic kidney disease. It must be noted, that both CK serum activity and serum creatinine concentration are both results of muscle breakdown, therefore there is a risk of multicollinearity. As the data were obtained from hospitals information systems, it was not possible to assure consistency of CK activity and creatinine concentration analytical methods. Information on agents used was based largely on patient self-report with laboratory confirmation in a minority; it should be noted however, that work by Liakoni et al. has shown that patient self-report is reliable particularly for traditional illicit drugs [48]. The data entered into the data collection tool were taken from medical histories and it was difficult to differentiate reported drug use from analytically confirmed in post-hoc analysis.

Due to revision of gathered dataset variables, we do not have information on CK activities and creatinine concentrations from EuroDEN presentations after first 12 months of the study.

## Conclusions

Abnormal values of CK activity may occur in up to half of presentations to the Emergency Department following acute recreational drug toxicity whose clinical signs suggest muscle

injury but severe rhabdomyolysis is much less common. Patients with rhabdomyolysis are at significantly higher risk of kidney injury (3.97 times higher, and 22.9 times higer for severe rhabdomyolysis) and spend more time in the hospital with a length of stay that is directly proportional to CK activity. Therefore, we suggest that it is valuable for patient management to check serum CK activity and creatinine concentrations in patients presenting to the Emergency Department with severe acute recreational drug toxicity as a part of routine laboratory testing.

## Acknowledgments

### Euro-DEN research group members

Lucie Chevillard, Florian Eyer, Miguel Galicia, Catalina Homar, Gesche Jürgens, Evangelia Liakoni, ME Liechti, Bruno Mégarbane, Oscar Miro, Adrian Moughty, Niall O'Connor, Raido Paasma, Kristiina Põld, Odd Martin Vallersnes, WS Waring.

## Author Contributions

**Conceptualization:** Wojciech Waldman, Jacek Sein Anand.

**Data curation:** Alison M. Dines.

**Formal analysis:** Piotr M. Kabata.

**Funding acquisition:** Paul I. Dargan.

**Investigation:** Paul I. Dargan, Jacek Sein Anand.

**Project administration:** Alison M. Dines.

**Software:** Piotr M. Kabata.

**Supervision:** David M. Wood, Paul I. Dargan.

**Validation:** David M. Wood, Paul I. Dargan, Jacek Sein Anand.

**Visualization:** Piotr M. Kabata.

**Writing – original draft:** Wojciech Waldman, Piotr M. Kabata.

**Writing – review & editing:** Wojciech Waldman, David M. Wood, Christopher Yates, Fridtjof Heyerdahl, Knut Erik Hovda, Isabelle Giraudon, Paul I. Dargan, Jacek Sein Anand.

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
