## [Decision Letter · Decision Letter 0]

12 Aug 2020

PONE-D-20-18319

Rhabdomyolysis related to acute recreational drug toxicity – a Euro-DEN study.

PLOS ONE

Dear Dr. Kabata,

Thank you for submitting your manuscript to PLOS ONE. After careful consideration, we feel that it has merit but does not fully meet PLOS ONE’s publication criteria as it currently stands. Therefore, we invite you to submit a revised version of the manuscript that addresses the points raised during the review process.

I have received the comments of the reviewers on your manuscript. The specific comments of the reviewers are included below. Please provide point by point response in your revised manuscript.

We look forward to receiving your revised manuscript.

Kind regards,

Muhammad Adrish

Academic Editor

PLOS ONE

Journal Requirements:

3. One of the noted authors is a group or consortium [Euro-DEN Research Group]. In addition to naming the author group, please list the individual authors and affiliations within this group in the acknowledgments section of your manuscript. Please also indicate clearly a lead author for this group along with a contact email address.

4. Your ethics statement must appear in the Methods section of your manuscript. If your ethics statement is written in any section besides the Methods, please move it to the Methods section and delete it from any other section. Please also ensure that your ethics statement is included in your manuscript, as the ethics section of your online submission will not be published alongside your manuscript.

5. Please include a copy of Table 2 which you refer to in your text (line 191).

Reviewers' comments:

Reviewer's Responses to Questions

**Comments to the Author**

1. Is the manuscript technically sound, and do the data support the conclusions?

Reviewer #1: Yes

2. Has the statistical analysis been performed appropriately and rigorously? 

Reviewer #1: No

3. Have the authors made all data underlying the findings in their manuscript fully available?

Reviewer #1: Yes

4. Is the manuscript presented in an intelligible fashion and written in standard English?

Reviewer #1: Yes

5. Review Comments to the Author

Reviewer #1: In this retrospective analysis, the authors described cases of rhabdomyolysis in patients who presnted to the Emergency Department with severe acute recreational drug toxicity. The study presents an interesting perspective on the frequency of rhabdomyolysis related to different drugs of abuse. However, the manuscript will require a major revision to make it suitable for publication. Areas that require the authors’ attention are itemized below.

Method

1. Page 10 line 129-Any justification for restricting the retrospective study to Oct 2013 to Sept 2014? Are there more recent data?

2. Page 11, line 165-168-Statistical analysis- The use of range and interquartile range as measures of data spread suggest that the data were not normally distributed. Therefore, the use of parametric test such as pearson's moment correlation and ANOVA for analysis may not be appropriated. The authors should consider the use of equivalent non parametric test for the analysis.

Results

1. Page 12, line 177-192- Correlation between CK and creatinine. This analysis is not necessary, as both biochemical parameter are the result of massive breakdown of skeletal muscles which occurs in Rhabdomyolysis. Early detection of acute renal failure can be achieved by monitoring serum creatinine and serum creatine kinase. CK is a better predictor of ARF due to rhabdomyolysis than creatinine. Since CK and creatinine are usually highly correlated in rhabsomyolysis, an attempt to find the correlation between them in this study will result in multicollinearity.

2. The result tables are not properly labelled. Also, the table do not present a clear distinction between the exposure and outcome variables making them difficult to interpret.

Discussion

1. The discussion needs to be better organized. There should be a clear pattern on how current findings connect to, support, elaborate on, or challenge conclusions of earlier studies.

Others are highlighted on the attached manuscript.

6. PLOS authors have the option to publish the peer review history of their article (what does this mean?). If published, this will include your full peer review and any attached files.

Reviewer #1: **Yes: **ISAAC OKOH ABAH

---

## [Author Response · Author response to Decision Letter 0]

14 Dec 2020

Dear Reviewers,

we would like to thank you for your review of our manuscript. We hope, that you will find our responses and revisions of the document satisfactory to reconsider the publication of our paper.

The style has been adjusted. 

2. We note that you have indicated that data from this study are available upon request. PLOS only allows data to be

available upon request if there are legal or ethical restrictions on sharing data publicly. For information on unacceptable

data access restrictions, please see http://journals.plos.org/plosone/s/data-availability#loc-unacceptable-data-access-

restrictions.

a) If there are ethical or legal restrictions on sharing a de-identified data set, please explain them in detail (e.g., data

contain potentially identifying or sensitive patient information) and who has imposed them (e.g., an ethics committee).

Please also provide contact information for a data access committee, ethics committee, or other institutional body to which

data requests may be sent.

The dataset contains date and time of patients admission to the hospital, which could potentially identify a patient; furthermore recreational drug use is a sensitive area for many individuals and therefore the risks associated with identification are greater in this context. The data can be obtained from European Drug Emergencies Network steering committee. 

The restrictions have been imposed by the EuroDEN steering committee.

For inquiries regarding data access, please contact Mrs Barbara Whetton, phone number: +44 (0)20 7188 5848 barbara.whetton@gstt.nhs.uk

b) If there are no restrictions, please upload the minimal anonymized data set necessary to replicate your study findings

as either Supporting Information files or to a stable, public repository and provide us with the relevant URLs, DOIs, or

accession numbers. Please see http://www.bmj.com/content/340/bmj.c181.long for guidelines on how to de-identify and

prepare clinical data for publication. For a list of acceptable repositories, please see

http://journals.plos.org/plosone/s/data-availability#loc-recommended-repositories.

3. One of the noted authors is a group or consortium [Euro-DEN Research Group]. In addition to naming the author group,

please list the individual authors and affiliations within this group in the acknowledgments section of your manuscript.

Please also indicate clearly a lead author for this group along with a contact email address.

Corrected

4. Your ethics statement must appear in the Methods section of your manuscript. If your ethics statement is written in any

section besides the Methods, please move it to the Methods section and delete it from any other section. Please also

ensure that your ethics statement is included in your manuscript, as the ethics section of your online submission will not

be published alongside your manuscript.

Corrected

5. Please include a copy of Table 2 which you refer to in your text (line 191).

Added

Reviewers' comments:

Reviewer #1: In this retrospective analysis, the authors described cases of rhabdomyolysis in patients who presnted to the Emergency Department with severe acute recreational drug toxicity. The study presents an interesting perspective on the frequency of rhabdomyolysis related to different drugs of abuse. However, the manuscript will require a major revision to make it suitable for publication. Areas that require the authors’ attention are itemized below.

Method

1. Page 10 line 129-Any justification for restricting the retrospective study to Oct 2013 to Sept 2014? Are there more

recent data?

Whilst Euro-DEN continued after Sept 2014 as the Euro-DEN Plus project creatinine concentration and CK activity are no longer collected therefore the data included in this study are only available for the period Oct 2013 – Sept 2014. We have clarification of this in the Materials and Methods.

2. Page 11, line 165-168-Statistical analysis- The use of range and interquartile range as measures of data spread suggest that the data were not normally distributed. Therefore, the use of parametric test such as pearson's moment correlation and ANOVA for analysis may not be appropriated. The authors should consider the use of equivalent non parametric test for the analysis.

Thank you – we agree, we have adjusted the statistical tests for non-normally distributed data. 

Results

1. Page 12, line 177-192- Correlation between CK and creatinine. This analysis is not necessary, as both biochemical parameter are the result of massive breakdown of skeletal muscles which occurs in Rhabdomyolysis. Early detection of acute renal failure can be achieved by monitoring serum creatinine and serum creatine kinase. CK is a better predictor of ARF due to rhabdomyolysis than creatinine. Since CK and creatinine are usually highly correlated in rhabsomyolysis, an attempt to find the correlation between them in this study will result in multicollinearity.

We are aware of the high correlation between CK activity and creatinine concentration. However our clinical experience has shown, that some of patients were have very high CK activity without significant kidney injury. The serum creatinine concentration in this study was used as the marker of kidney injury. Unfortunately, due to study design (patients were treated mainly in emergency departments and only the highest recorded creatinine concentration was recorded in the dataset) we could not apply KDIGO criteria for acute kidney injury. We have addressed this in revised limitations of the paper.

2. The result tables are not properly labelled. Also, the table do not present a clear distinction between the exposure and outcome variables making them difficult to interpret.

Thank you - we corrected the labelling of the tables. Hopefully, this makes them more comprehensible.

Discussion

1. The discussion needs to be better organized. There should be a clear pattern on how current findings connect to, support, elaborate on, or challenge conclusions of earlier studies.

Thank you, we shortened the discussion and added connections to our study, to make it more clear and concise.

Thank you for consideration of our paper, and we hope that the revised version presents a higher standard, that would be suitable for publication.

Best regards

Piotr Maciej Kabata, on behalf of Euro-DEN Research Group

---

## [Decision Letter · Decision Letter 1]

18 Jan 2021

Rhabdomyolysis related to acute recreational drug toxicity – a Euro-DEN study.

PONE-D-20-18319R1

Dear Dr. Kabata,

We’re pleased to inform you that your manuscript has been judged scientifically suitable for publication and will be formally accepted for publication once it meets all outstanding technical requirements.

Kind regards,

Muhammad Adrish

Academic Editor

PLOS ONE

Additional Editor Comments (optional):

All recommended edits have been addressed.

Reviewers' comments:

Reviewer's Responses to Questions

**Comments to the Author**

1. If the authors have adequately addressed your comments raised in a previous round of review and you feel that this manuscript is now acceptable for publication, you may indicate that here to bypass the “Comments to the Author” section, enter your conflict of interest statement in the “Confidential to Editor” section, and submit your "Accept" recommendation.

Reviewer #1: (No Response)

2. Is the manuscript technically sound, and do the data support the conclusions?

Reviewer #1: Yes

3. Has the statistical analysis been performed appropriately and rigorously? 

Reviewer #1: Yes

4. Have the authors made all data underlying the findings in their manuscript fully available?

Reviewer #1: Yes

5. Is the manuscript presented in an intelligible fashion and written in standard English?

Reviewer #1: No

6. Review Comments to the Author

Reviewer #1: The authors have sufficiently addressed the issues raised in the previous review, except that columns heading are missing in Table 1. Authors should ensure that Tables are formatted in line with the journal's style. For instance, the table titles should be at the top of the table and not at the bottom.

7. PLOS authors have the option to publish the peer review history of their article (what does this mean?). If published, this will include your full peer review and any attached files.

Reviewer #1: **Yes: **ISAAC OKOH ABAH

---

## [Editor Report · Acceptance letter]

25 Jan 2021

PONE-D-20-18319R1 

Rhabdomyolysis related to acute recreational drug toxicity – a Euro-DEN study. 

Dear Dr. Kabata:

I'm pleased to inform you that your manuscript has been deemed suitable for publication in PLOS ONE. Congratulations! Your manuscript is now with our production department. 

Kind regards, 

on behalf of

Dr. Muhammad Adrish 

Academic Editor

PLOS ONE